# Effects of Focused Ion Beam Lithography on La$_{2-x}$Sr$_x$CuO$_4$ Single Crystals

Roberta Caruso [1], Fernando Camino [1], Genda Gu [1], John M. Tranquada [1], Myung-Geun Han [1], Yimei Zhu [1], Anthony T. Bollinger [1] and Ivan Božović [1,2,*]

1   Brookhaven National Laboratory, Upton, NY 11973-5000, USA
2   Department of Chemistry, Yale University, New Haven, CT 06520, USA
*   Correspondence: bozovic@bnl.gov

**Abstract:** Focused ion beam (FIB) milling is a mask-free lithography technique that allows the precise shaping of 3D materials on the micron and sub-micron scale. The recent discovery of electronic nematicity in La$_{2-x}$Sr$_x$CuO$_4$ (LSCO) thin films triggered the search for the same phenomenon in bulk LSCO crystals. With this motivation, we have systematically explored FIB patterning of bulk LSCO crystals into micro-devices suitable for longitudinal and transverse resistivity measurements. We found that several detrimental factors can affect the result, ultimately compromising the possibility of effectively using FIB milling to fabricate sub-micrometer LSCO devices, especially in the underdoped regime.

**Keywords:** superconductivity; lithography; focused ion beam

## 1. Introduction

The discovery of electronic nematicity in single-crystal LSCO films fabricated by molecular-beam epitaxy at all doping levels has attracted much attention [1,2]. Still, it also triggered the question of whether this behavior is perhaps peculiar to thin films—e.g., due to the epitaxial strain, gradients in the film thickness or composition, or to lithography, since any of these could break the (tetragonal) rotational symmetry—or if it is intrinsic and generic to LSCO (and other cuprates). A previous work using thin films employed a specifically designed lithographic pattern to measure the angle-resolved transverse resistivity (ARTR) based on the consideration that in an isotropic material, no voltage should be detected in the direction orthogonal to the current flow direction in the absence of a magnetic field. In contrast, in anisotropic (e.g., orthorhombic) materials, a transverse voltage is detected [1]. It is an open question as to whether the same electronic symmetry-breaking can also be observed in bulk LSCO single crystals. The problem here is that the typical angular variations of longitudinal resistivity are fairly minor, of the order of 1%. Hence, standard millimeter-size LSCO crystals pose a challenge because the low resistivity and the sample thickness make the total resistance extremely small and measuring variations of 1% or lower with sufficient accuracy is difficult.

For this reason, it is necessary to chisel micrometer and sub-micrometer-size devices out of bulk LSCO single crystals. Focused ion beam (FIB) milling has enabled important discoveries in different materials [3–7]. Several groups reported the fabrication of cuprate-based devices using Bi$_2$Sr$_2$CaCu$_2$O$_{8+x}$ and YBa$_2$Cu$_3$O$_7$ [5,6]. These results prompted us to use this technique to probe electronic nematicity in bulk LSCO single crystals. For this purpose, it is best to study underdoped LSCO because the observed anisotropy of in-plane transport is more significant in the lower doping region [1,2]. Thus, we used FIB milling to carve the bulk single crystals of LSCO with $x = 0.07$, i.e., La$_{1.93}$Sr$_{0.07}$CuO$_4$, into Hall bars of different sizes and shapes. We employed a range of strategies to reduce sample contamination by the FIB process steps. Still, we were not able to observe a

superconducting transition down to the temperature $T = 4$ K. In principle, using LSCO crystals with the Sr content closer to optimal doping could allow the observation of a superconducting transition in FIB-patterned devices, but given their small resistance, it is not clear whether these devices will be suitable to demonstrate electronic nematicity in 3D crystals unambiguously.

## 2. Results

### 2.1. Devices

We fabricated several LSCO devices with different form factors. In Figure 1a, we show an example of the first series of devices, shaped as 'classic' Hall bars with well-defined current and voltage contacts. The device thickness is approximately 1 μm. In Figure 1b, we offer the latest batch of devices that were not shaped as Hall bars; instead, the lamella was slimmed down and the Pt contacts were deposited at its periphery. In this case, the device was thinned down to approximately 500 nm. The rationale for this device redesign is that FIB milling, especially at near-normal incident angles, causes much more damage than FIB at grazing angles. Thus, by reducing the FIB processing, we also expect to reduce beam-induced damage. In this second generation of samples, we also introduced two cleaning steps, at 15 keV and 8 keV, at grazing angles. FIB milling relies on the scanning-electron microscopy (SEM) imaging of the sample, and the electron beam can cause damage to the material or contaminate the surface with carbon deposition from hydrocarbons present in the chamber [8]. To reduce this, we used a much lower SEM electron current.

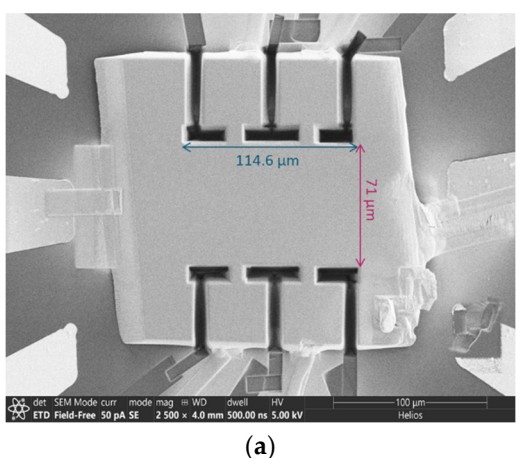
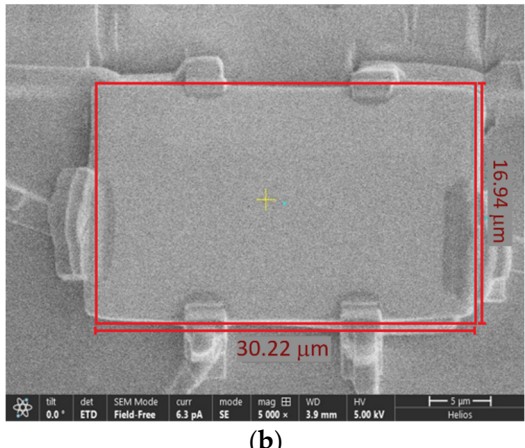

(**a**)  (**b**)

**Figure 1.** (**a**) 'First generation' device, shaped as a classic Hall bar. (**b**) 'Second generation' device, with shaping reduced to the minimum.

### 2.2. Transport Measurements

In each sample, we measured both the longitudinal and the transverse resistance at the same time. In Figure 2, we show the results of measurements for the two samples in Figure 1. Neither sample fully turned superconducting down to $T = 4$ K. For sample 1 (Figure 2a), the longitudinal resistance shows an onset of a very broad transition, decreasing by two orders of magnitude but not yet vanishing at 4 K. In the same sample, the transverse resistance shows a distinct upturn starting at $T \approx 16$ K, possibly caused by superconducting fluctuations in analogy with the observation of a sharp peak before the superconducting transition observed in thin films [1]. For sample 2, longitudinal and transverse resistance decrease at low temperatures, although this is not as pronounced as in sample 1. No upturn in the transverse resistance is observed in sample 2, so the feature observed in sample 1 is more likely related to sample inhomogeneities rather than to superconducting fluctuations.

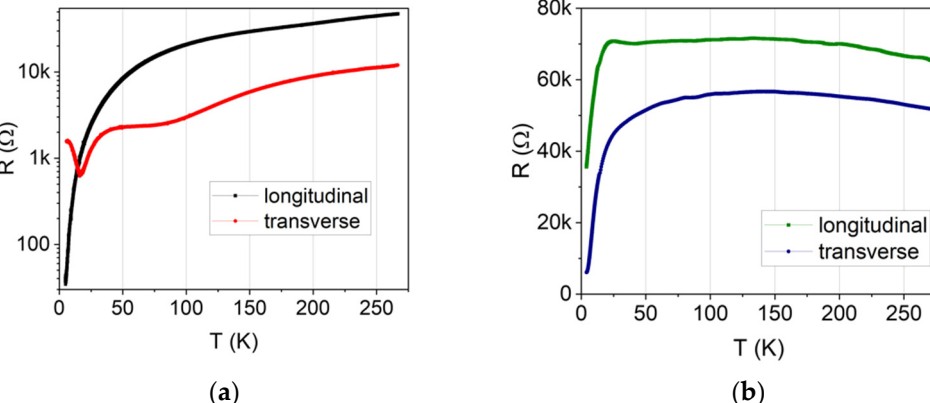

**Figure 2.** The temperature dependence of resistance in (**a**) the sample in Figure 1a and (**b**) the sample in Figure 1b.

### 2.3. TEM Characterization

We performed a structural and chemical TEM analysis, as shown in Figure 3. The cross-sectional TEM sample was prepared by a focused ion beam (FIB) with 2 keV Ga ions for the final milling. The high-angle annular dark-field (HAADF) scanning TEM (STEM) images (Figure 3a–c) show an amorphous layer (~17 nm thick) and crystalline $La_{2-x}Sr_xCuO_4$. A chemical analysis was performed on O K-edge, La M-edge, and Ga L-edge based on electron energy loss spectroscopy (EELS). Our results show that there is a significant Ga contamination in the amorphous layer, where the La and O signals are also lower compared to those in the crystalline area. In the crystalline area, all signals for the three elements are uniform. The Ga L-edge signal in the crystalline area may come from contamination due to the lithography steps necessary to shape the device, or it could be due to the top and bottom surfaces of the cross-sectional TEM samples, which are processed by the Ga-ion milling. In any case, the use of FIB technique unambiguously causes Ga contamination deep in LSCO single crystals.

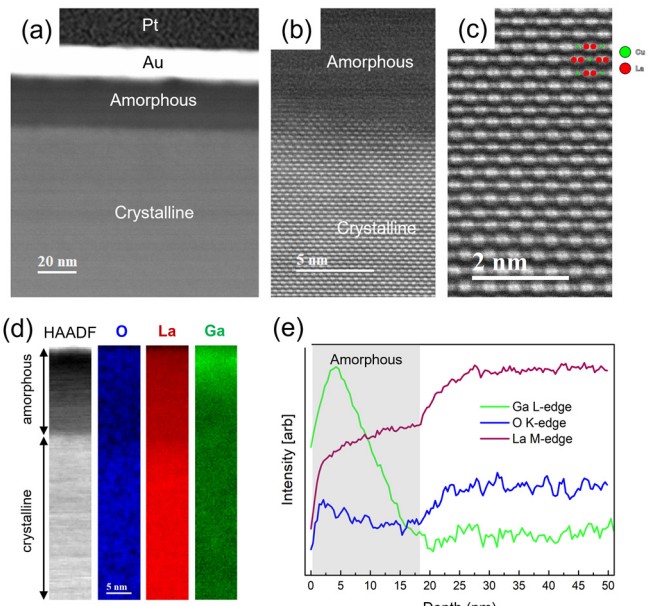

**Figure 3.** Structural and chemical analysis of the cross-sectional sample. (**a**–**c**) HAADF STEM images taken along the [011] direction. The Pt and Au layers were deposited before the FIB milling to protect

the sample. An amorphous layer (~17 nm thick) is observed. (**d**) EELS elemental mapping showing spatial distributions of O K-edge, La M-edge, and Ga L-edge signals. (**e**) Line profiles from elemental maps. The amorphous layer shows a significant Ga signal while La and O signals are low.

## 3. Discussion

Many factors can contribute to the suppression of the superconducting transition. The main factor to address is Ga contamination during the ion milling process, which has two significant effects: forming an amorphous layer next to the FIB-cut surface of the sample and creating structural defects deep inside the device under study [9–11].

When FIB milling is performed at the typical energy of 30 keV and a grazing angle between 0° and 5° with respect to the surface, the amorphous layer is usually 20–30 nm thick, and this thickness is independent of the beam current and the exposure time. However, when milling is performed at incident angles above 40° with respect to the surface, the amorphous layer thickness increases significantly and becomes current and time-dependent—it can reach 100 nm.

A common approach is to use FIB at a low grazing angle and then to further reduce the amorphous layer thickness from 20–30 nm down to 3–5 nm by adding one or more cleaning steps at low energies (typically 15 keV and 8 keV) at grazing angles [7,9]. In our case, this method provided only a partial solution to Ga contamination, since a significant portion of the shaping must occur at steep incident angles. We tried gentle ex-situ Ar milling to remove the amorphized layer, but this proved detrimental to the sample. We then tried to reduce contamination by minimizing the shaping steps and obtained a slightly better result (see Figure 2).

At high beam energies, Ga ions have a long penetration depth, thus causing implantation and atomic displacement deep inside the material. Ga ions are also highly reactive and prone to forming compounds with other elements within the sample. In LSCO and other cuprates, Ga ions will likely react with oxygen ions. Gallium oxides, $Ga_2O_3$ and $Ga_2O$, are stable compounds that are unlikely to be removed by thermal annealing [10,11]. Gallium contamination at the 1% level was shown to reduce $T_c$ of LSCO significantly, and the effect is more pronounced at lower doping values [12]. In typical FIB processes, $Ga^{2+}$ contamination is estimated to be between 4% and 9% from calculations, and between 0.2% and 1.8% in carefully designed experiments, for grazing angle milling [9]. We annealed one of the samples under high pressure of $O_2$ and another in $O_3$, both at 400 °C, but neither improved their superconducting properties. At higher annealing temperatures, the material deteriorates irreversibly.

Another significant source of contamination comes from the in-situ Pt contact deposition. In FIB systems, contacts are fabricated using ion-assisted chemical vapor deposition (IA-CVD). A volatile metallo-organic precursor is injected through a nozzle and adsorbed on the sample surface. Then, Ga ions impacting the surface evaporate the volatile part of the compound while the metal is deposited on the sample. The contacts deposited using this technique usually have a lower conductivity compared to the same metals deposited with other methods and often show non-linear behavior [7]. Aside from this issue, the use of IA-CVD contributes to contamination with Pt atoms that can penetrate the device and cause other defects and dislocations in addition to the ones caused by Ga ions. We successfully optimized the contact deposition parameters by reducing the ion-beam energy down to 16 keV, as well as the beam current to 12pA, to minimize the formation of Schottky barriers that could affect the transport measurements on the LSCO Hall bar devices. In Figure 4, we show the resistance versus frequency dependence for two different devices, before and after the optimization of parameter of Pt deposition. In Figure 4a, we show the data from a device that was not patterned (and discarded as not useful for further analysis), while in Figure 4b we show the resistance versus frequency data for the sample shown in Figure 1b. In this case, the resistance does not depend upon the frequency up to about 1 kHz, confirming the presence of an ohmic contact. However, our efforts to minimize Pt

contamination so far yielded only marginal improvements, in the sense that regardless of this optimization we did not observe a complete superconducting transition.

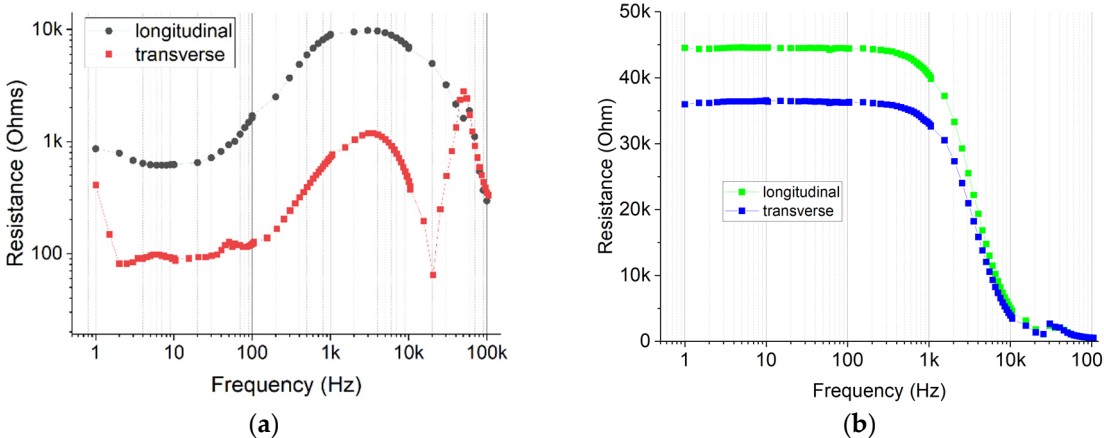

**Figure 4.** (**a**) Resistance versus frequency for the first iteration of unpatterned LSCO devices. (**b**) resistance versus frequency curve for the sample shown in Figure 1b.

Another source of defects can be damage caused by SEM imaging. This was previously reported as degrading the superconducting properties of LSCO nanowires and thus should not be used [13]. However, it is virtually impossible to avoid using SEM imaging in FIB-SEM systems, so our approach has been to use the lowest possible electron beam energy and current to image our device during FIB milling.

Finally, one should consider possible changes in the sample stoichiometry induced by the ion beam. During FIB processes, elements with lower sublimation temperatures will preferentially leave the sample. In the case of LSCO, oxygen has the lowest sublimation temperature. It thus will be the first to leave the sample, causing oxygen deficiency that can further weaken the superconducting state.

## 4. Materials and Methods

A high-quality bulk LSCO single crystal with $x = 0.07$ was grown by using the floating zone technique at Brookhaven National Laboratory. It was then aligned by Laüe diffraction and manually polished so that the *a-b* planes were exposed and parallel to the surface. The typical accuracy in this process is 1–2°. The flake is then mounted on a vertical sample holder and loaded into a Helios 5 UX DualBeam from ThermoFisher Scientific (Waltham, MA, USA), using Ga ions as a FIB source and equipped with an SEM for sample imaging. The fabrication process is depicted in Figure 5. First, the surface of the crystal was polished using a 30 keV ion beam at a grazing angle (Figure 5a), then the ion beam was used to dig a trench to expose the lamella, which was used to realize the device (Figure 5b). The lamella was then cut using FIB, extracted using an in-situ micromanipulator (Figure 5c,d), attached to a vertical holder, and polished at a grazing angle using low energy FIB. Finally, it was attached to a pre-patterned $LaSrAlO_4$ substrate using in-situ ion-assisted Pt deposition (Figure 5e). Since the $LaSrAlO_4$ substrate is insulating, a thin (8 nm) gold film was evaporated on the surface to ensure proper grounding and stable FIB-SEM processing. Finally, the lamella was shaped using FIB and Pt contacts were made using FIB-assisted deposition (Figure 5f). The shaping and contact deposition were performed at 52° where possible or at 90° with respect to the sample surface.

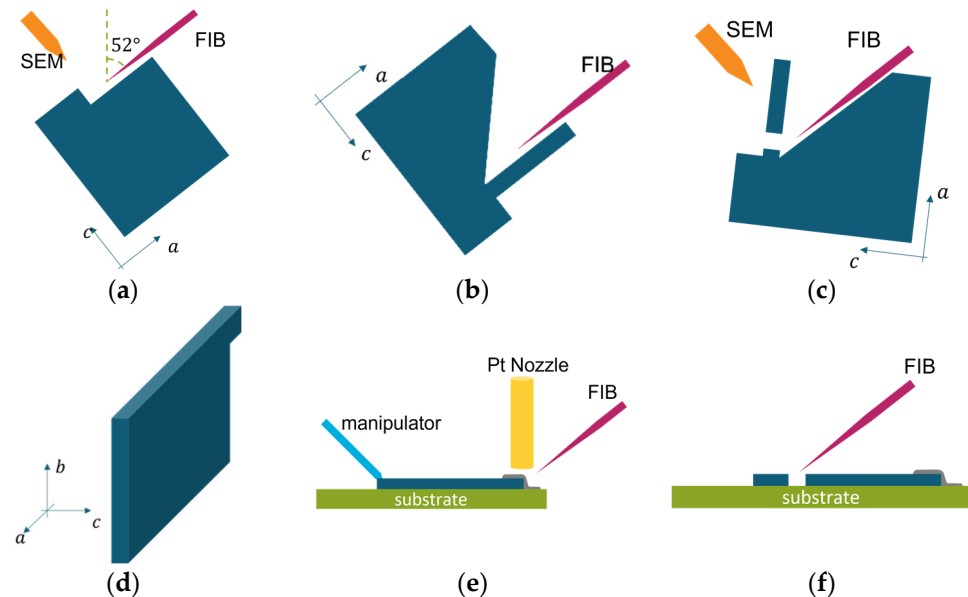

**Figure 5.** Device fabrication process. (**a**) The outer-facing side of the lamella was polished using a high-energy FIB. (**b**) The back side of the lamella was exposed by digging a trench with a high-energy FIB. (**c**) The lamella was freed with a U-cut around the perimeter. (**d**) The lamella was then attached to a vertical holder and polished at a grazing angle using a low-energy FIB. (**e**) The lamella was placed flat on the substrate and contacted with Pt in-situ deposition. (**f**) The device was shaped, and the Pt contacts were deposited.

## 5. Conclusions

Our attempts to manufacture Hall bar devices out of bulk crystals of underdoped LSCO using FIB have proven to be quite challenging. There are many sources of contamination that are hard to control with the necessary precision to achieve satisfactory results. We solved some of the issues, but so far, we have been unable to obtain a device in which a complete superconducting transition is observed above 4 K. Despite several reports of successful FIB patterning of other cuprates, we conclude that for LSCO, a new and different approach may be necessary. The first step towards a different approach would be to use much thinner samples, around 100 nm–200 nm, closer to the typical thickness of samples used in tunnel electron microscopy (TEM). In fact, performing the U-cut in Figure 5c requires the use of a high-energy Ga-ion beam at incident angles for extended times (around 60 s to 90 s), which is detrimental for the sample. A thinner lamella allows the use of a lower energy Ga ion beam for a shorter time, reducing the exposure and the subsequent contamination of the sample. In the future, therefore, we would consider using a vertical sample holder instead of a standard horizontal one. This configuration would allow the Pt contact deposition at grazing angles, thus reducing Ga contamination from perpendicular milling. Finally, we would consider using FIB with other ions, as described for instance in [14], with Xenon in particular, since it is far less reactive and also heavier than Ga. These two characteristics may enable the fabrication of devices that are better, larger, and more suitable for transport measurements.

**Author Contributions:** G.G. synthesized the LSCO crystals. R.C. and F.C. performed FIB patterning. R.C. performed transport measurements. M.-G.H. and Y.Z. performed the TEM analysis. J.M.T. and A.T.B. contributed to the manuscript. I.B. conceived and supervised the project. All authors have read and agreed to the published version of the manuscript.

**Funding:** This research was supported by the DOE, Basic Energy Sciences, Materials Sciences, and Engineering Division. This research also used resources from the Center for Functional Nanomaterials, a U.S. DOE Office of Science Facility at Brookhaven National Laboratory under Contract No. DE-SC0012704.

**Data Availability Statement:** All data are available upon request.

**Conflicts of Interest:** The authors declare no conflict of interest.

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
