# Peer review of "Effects of Focused Ion Beam Lithography on La2−xSrxCuO4 Single Crystals"

_condensedmatter, doi:10.3390/condmat8020035_

Round 1
Reviewer 1 Report
In this manuscript, authors reported a well-designed investigation on the effectiveness of FIB milling in the LSCO crystal. The main purpose of such an investigation is to study the electronic nematicity in this archetypical high transition temperature superconductor. A well controlled the sample will help to clarify the ambiguity in previous research. Scientifically, this study is of great current interest in the field of superconductivity and the experiment is properly designed. Two iterations of the FIB procedures were reported here that reveal the possible sources of the deterioration of superconductivity in the lamella. I like the candid attitude the authors expressed here that, though these lamellae are not suitable for the electronic nematicity study, authors carefully studied all factors which may cause the degrade of the sample quality. This negative result is very helpful to the scientific community in future material preparation. In this sense, I’d like to suggest a publication this manuscript with several minor revisions as listed below.
1. In the line 41, “more significant the lower the doping is (1,2).” should be “more significant in lower doping region (1,2).” or something similar.
2. In the Fig. 1(b), the cyan color used to mark the sample geometry and dimension is hard to read. Can authors change that color to something with high contrast?
3. For the lamellae reported in Fig. 1(a) and (b), can authors include the thickness information in the main text?
Author Response
C1. In this manuscript, authors reported a well-designed investigation on the effectiveness of FIB milling in the LSCO crystal. The main purpose of such an investigation is to study the electronic nematicity in this archetypical high transition temperature superconductor. A well controlled the sample will help to clarify the ambiguity in previous research. Scientifically, this study is of great current interest in the field of superconductivity and the experiment is properly designed. Two iterations of the FIB procedures were reported here that reveal the possible sources of the deterioration of superconductivity in the lamella. I like the candid attitude the authors expressed here that, though these lamellae are not suitable for the electronic nematicity study, authors carefully studied all factors which may cause the degrade of the sample quality. This negative result is very helpful to the scientific community in future material preparation. In this sense, I’d like to suggest a publication this manuscript with several minor revisions as listed below.
A1. We are grateful to the Review for the positive comments, the appreciation of our candidly reporting ‘negative’ result, and the recommendation for the paper to be published in Condensed Matter.
C2. In the line 41, “more significant the lower the doping is (1,2).” should be “more significant in lower doping region (1,2).” or something similar.
A2. We changed the text in line 41-42. The new sentence reads: “the observed anisotropy of in-plane transport is more significant in the lower doping region.”
C3. In the Fig. 1(b), the cyan color used to mark the sample geometry and dimension is hard to read. Can authors change that color to something with high contrast?
A3. We modified Fig 1b so that now the markers have a higher contrast.
C4. For the lamellae reported in Fig. 1(a) and (b), can authors include the thickness information in the main text?
A4. We added the information about the thickness on line 55 and 57. The new paragraph reads: “In Fig. 1a, we show an example of the first series of devices, shaped as ‘classic’ Hall bars with well-defined current and voltage contacts. The device thickness is approximately 1 µm. In Fig. 1b, we offer the latest batch of devices that were not shaped as Hall bars; instead, the lamella was just slimmed down, and the Pt contacts were deposited at its periphery. In this case, the device is thinned down to approximately 500 nm.
Reviewer 2 Report
I was excited when I was invited to review the paper "Effects of Focused Ion Beam lithography on La2-xSrxCuO4 single crystals" by R. Caruso et al., as I am very interested in novel fabrication methods of complex materials. With all due respect to my colleagues, who I value greatly, I cannot give this paper a positive review. In this paper, two failed attempts of Focused Ion Beam fabrication akin to a huge TEM-lamella process of underdoped LSCO crystals and their failed transport data is presented. The discussion section contains nothing but generic speculations about why the process may have failed. There is no rigorous analysis at all to corroborate or at least investigate these potential sources of failure. If a FIB-induced amorphization is to blame, with potential Ga-reaction, detailed TEM studies should be shown that quantify the extent of amorphization and Ga-implantation. Why does this effect persist after the low-V Ar milling (again, TEM is needed to show how the layer has changed). No chemical information about the surface at all is given, and no information about the possibility of superconductivity in the bulk is shown. As the authors state, many other cuprates with much more mobile dopants (oxygen) have been very successfully treated with Ga-FIBs in the past. It is at least very counter-intuitive that the more strongly bound dopants, La/Sr, would massively vary their composition over many microns, while oxygen in hole-doped cuprates is known to stay in place (as in the good old mesa experiments for example).
It is much more likely that the authors encounter a three-shell issue that originates in their slab method. If the amorphous layer is chemcially reactive, especially with oxides, surface chemisty can further seggregate the FIB-damage layer. Most likely this happens through the loss of oxygen at the outermost surface. In this case, a poorly conductive but metallic Cu-rich layer forms as the outer shell, the inner shell being a thin but strong ceramic insulator, coating a perfectly fine LSCO crystal. The currents then flow in the outer shell, and only benign dirty-metal physics is probed using surface contacts. This is a speculation, but likely a detailed TEM analysis will bring the authors to that conclusion. Furthermore, I am convinced that any magnetic probe (Hall bar, SQUID, cantilever magnetometry) will uncover the slab to be superconducting. It would be a surprise if indeed superconductivity was lost given the thickness of the slab used.
All in all, and I mean this in a very collegial and supportive way, I wonder what the authors main take home message of this paper is? It reads like a labbook of two failed attempts, so what does the reader learn about the "Effects of Focused Ion Beam lithography on La2-xSrxCuO4 single crystals"? This could be a very interesting story, even as a negative result on the nematicity itself, if the effects were actually demonstrated and quantified. What is really going on, why does this not work?
Author Response
C1. I was excited when I was invited to review the paper "Effects of Focused Ion Beam lithography on La2-xSrxCuO4 single crystals" by R. Caruso et al., as I am very interested in novel fabrication methods of complex materials. With all due respect to my colleagues, who I value greatly, I cannot give this paper a positive review. In this paper, two failed attempts of Focused Ion Beam fabrication akin to a huge TEM-lamella process of underdoped LSCO crystals and their failed transport data is presented. The discussion section contains nothing but generic speculations about why the process may have failed.
A1. We are grateful to the Reviewer for a laudatory comment about the present authors, and for the candid assessment of the paper’s value. As for the latter, we would like to make two points. First, this is a Conference paper reporting the work in progress. Today, regular research papers are all published online and immediately available. Hence, the entire purpose of attending a research conference is to present and hear unpublished results and the work in progress, so this was indeed explicitly invited by the conference organizers and Special Issue Editors. The second point is that truthfully reporting negative results can be a valuable service to the research community, preventing future mistakes and erroneous conclusions, saving precious time, effort, and resources. Note that we employed the standard FIB protocols that many other users at the same and different national user facilities have been following. Hence, they could have had the same issues, probably without even realizing them. Once they read this, they may be compelled to revisit their data, papers, and conclusions — and, in particular, be mindful of this in the future.
C2. There is no rigorous analysis at all to corroborate or at least investigate these potential sources of failure. If a FIB-induced amorphization is to blame, with potential Ga-reaction, detailed TEM studies should be shown that quantify the extent of amorphization and Ga-implantation. Why does this effect persist after the low-V Ar milling (again, TEM is needed to show how the layer has changed).
A2. Following the Reviewers recommendation, we enlisted the help of one of the leading TEM groups in US, led by Yimei Zhu. They performed a detailed TEM characterization, including high-angle annular dark-field (HAADF) scanning TEM (STEM) imaging and a chemical analysis on O K-edge, La M-edge, and Ga L-edge, based on electron energy loss spectroscopy (EELS). M. G. Han and Y. Zhu are now added as coauthors, as their new TEM data inform our interpretation and strengthen our conclusions.
On page 3, lines 84-97, we have added the following new paragraph:
“We performed a structural and chemical TEM analysis, as shown in Fig. 3. The cross-sectional TEM sample was prepared by a focused ion beam (FIB) with 2 keV Ga ions for the final milling. The high-angle annular dark-field (HAADF) scanning TEM (STEM) images (Fig. 3a-c) show an amorphous layer (~ 17 nm thick) and crystalline La2-xSrxCuO4. A chemical analysis was performed on O K-edge, La M-edge, and Ga L-edge, based on electron energy loss spectroscopy (EELS). Our results show that there is a significant Ga contamination in the amorphous layer, where the La and O signals are also lower compared to those in the crystalline area. In the crystalline area, all signals for the three elements are uniform. The Ga L-edge signal in the crystalline area may come from contamination due to the lithography steps necessary to shape the device, or it can be due to the top and bottom surfaces of the cross-sectional TEM samples, which are processed by the Ga-ion milling. In any case, the use of FIB technique unambiguously causes Ga contamination deep in LSCO single crystals.”
We also added a new Figure 3 in the revised version of the manuscript. These experiments required significant new effort and time investment, and we hope this will be appreciated.
C3. No chemical information about the surface at all is given, and no information about the possibility of superconductivity in the bulk is shown.
A3. We have expanded our original manuscript including the HAADF images shown in fig. 3a and 3b of the manuscript and the electron energy loss spectroscopy (EELS) analysis in fig.3d and 3e, to obtain more information about the chemical composition of the devices. The results presented in the revised version of the manuscript show that the surface of the device is heavily contaminated with Ga ions, while La and O signals are suppressed in the same region. On the other hand, HAADF images shown in fig. 3a and 3b of the manuscript show the presence of an amorphous layer approximately 17nm thick, followed by a crystalline region down below the sample. The crystalline structure alone may suggest a superconducting behavior in the bulk, however the presence of the Ga signal from EELS analysis also in the bulk points towards a contamination that could potentially destroy the superconducting state. Note that the LSCO crystal used in this experiment lies in the low doping region of the phase diagram, where even small contaminations can wash out the superconducting state.
C4. As the authors state, many other cuprates with much more mobile dopants (oxygen) have been very successfully treated with Ga-FIBs in the past. It is at least very counter-intuitive that the more strongly bound dopants, La/Sr, would massively vary their composition over many microns, while oxygen in hole-doped cuprates is known to stay in place (as in the good old mesa experiments for example).
A4. Our findings raise concerns about at least some of the claims made in the literature by groups that used the same techniques and, in some cases, exactly the same equipment and protocols. For one, we are holding back ourselves on one potentially important paper already written. We patterned Fe(Te,Se) crystals using FIB at CFN-BNL using similar process protocol and parameters. in cryogenic magneto-transport measurements we observed signatures of electronic nematicity and quasi-1D superconducting fluctuations. The discovery of these new effects, if genuine and intrinsic, could have a strong impact in our field. But we are worried that some of the observed effects may not be intrinsic to Fe(Te,Se), but are rather FIB-induced. So, for now we are holding this report back until we clarify the question of the presence, extent, and role of FIB-induced damage.
C5. It is much more likely that the authors encounter a three-shell issue that originates in their slab method. If the amorphous layer is chemcially reactive, especially with oxides, surface chemisty can further seggregate the FIB-damage layer. Most likely this happens through the loss of oxygen at the outermost surface. In this case, a poorly conductive but metallic Cu-rich layer forms as the outer shell, the inner shell being a thin but strong ceramic insulator, coating a perfectly fine LSCO crystal. The currents then flow in the outer shell, and only benign dirty-metal physics is probed using surface contacts. This is a speculation, but likely a detailed TEM analysis will bring the authors to that conclusion.
A5. We value the three-shell toy model proposed by Reviewer, however the TEM analysis presented in the revised version of the manuscript did not show any sign of an intermediate, strong insulating layer between the amorphous, Ga-rich top layer and the crystalline bulk. Thus, the incomplete superconducting behavior that we observe is likely to extend to the bulk rather than to be limited to the surface. The Ga L-edge signal measured with EELS persists deep into the crystal, although low, possibly indicating the presence of this element well beyond the surface amorphous layer. We stress that the LSCO crystal used in this experiment is in the heavily underdoped regime, so even a small contamination (of the order of 1% or less) will have a more severe role in the suppression of the superconducting transition compared to other samples with optimal doping.
C6. Furthermore, I am convinced that any magnetic probe (Hall bar, SQUID, cantilever magnetometry) will uncover the slab to be superconducting. It would be a surprise if indeed superconductivity was lost given the thickness of the slab used.
A6. We value the hypothesis of the Reviewer, but since our TEM analysis did not show the presence of an intermediate insulating layer between the metallic surface and the superconducting bulk, we believe that if any superconducting behavior was present we should have been able to detect it with simple transport measurements. We can think of our device as constituted by two conductance channels in parallel, σ1 and σ2 one metallic (σ1) and one superconducting below a certain temperature (σ2). Once below the transition temperature, the superconducting channel will have a lower conductance, and thus should dominate in transport measurements. We only see an incipient transition in our devices, which indicates that below a certain temperature σ2 becomes dominant, but it remains finite.
C7. All in all, and I mean this in a very collegial and supportive way, I wonder what the authors main take home message of this paper is? It reads like a labbook of two failed attempts, so what does the reader learn about the "Effects of Focused Ion Beam lithography on La2-xSrxCuO4 single crystals"? This could be a very interesting story, even as a negative result on the nematicity itself, if the effects were actually demonstrated and quantified. What is really going on, why does this not work?
A7. We would like to reiterate that we used the standard FIB protocol at BNL CFN facility, that has been followed by numerous other users. At least they should hear about these issues, and perhaps re-evaluate their claims and conclusions. And the problems may well be more widespread.
Reviewer 3 Report
In this manuscript, the authors report their results on using focused ion beam (FIB) milling to fabricate LSCO micro-devices for studying electronic nematicity. The authors show the two devices fabricated with different geometry and using different FIB processing times, and present the transport measurements. Unfortunately, neither of the devices show vanishing resistance down to temperature of T=4K. The authors then discuss the possible reasons that make FIB detrimental to the LSCO superconductivity properties, including Ga and Pt contamination and possible damages caused by SEM imaging. Unfortunately, I am afraid I cannot recommend its publication in Condensed Matter for the following reasons:
1. In addition to merely reporting the negative results, the authors should propose potential or alternative methods to overcome the encountered challenges.
2. More details about the original bulk sample and its fabrication process should be provided.
(1) Where is the bulk LSCO sample purchased and what is the quality of the original sample before fabrication?
(2) On Page 4, the authors wrote “We successfully optimized the contact deposition parameters to avoid the formation of Schottky barriers that could affect the transport measurements on the LSCO Hall bar devices.” What are the deposition parameters and how bad the Schottky barriers look like? Data should be provided to support this successful optimization.
3. Given that the authors suspect Ga contamination is a main factor that suppresses the superconducting transition, the authors may want to consider FIB with other ions. People have explored FIB with ions other than Ga (for example, Liu, J., Niu, R., Gu, J., Cabral, M., Song, M. and Liao, X., 2020. Effect of ion irradiation introduced by focused ion-beam milling on the mechanical behaviour of sub-micron-sized samples. Scientific reports, 10(1), pp.1-8.)
Author Response
C1. In this manuscript, the authors report their results on using focused ion beam (FIB) milling to fabricate LSCO micro-devices for studying electronic nematicity. The authors show the two devices fabricated with different geometry and using different FIB processing times, and present the transport measurements. Unfortunately, neither of the devices show vanishing resistance down to temperature of T=4K. The authors then discuss the possible reasons that make FIB detrimental to the LSCO superconductivity properties, including Ga and Pt contamination and possible damages caused by SEM imaging.
A1. We thank the Reviewer for a fair assessment of the paper’s content.
C2. Unfortunately, I am afraid I cannot recommend its publication in Condensed Matter for the following reasons: In addition to merely reporting the negative results, the authors should propose potential or alternative methods to overcome the encountered challenges.
A2. We added our considerations on potential alternative methods for future developments on line 197. The new text reads as follows: “The first step towards a different approach would be to use much thinner samples, around 100 nm-200 nm, closer to the typical thickness of samples used in TEM studies. In fact, performing the U-cut in Figure 4c requires the use of a high-energy Ga-ion beam at incident angles for extended times (around 60 s to 90 s), which is detrimental for the sample. A thinner lamella allows the use of a lower-energy Ga-ion beam for a shorter time, reducing the exposure and the subsequent contamination of the sample. Then, for the future, we consider using a vertical sample holder instead of the standard horizontal one. This configuration allows the Pt contact deposition at grazing angles, thus reducing Ga contamination from perpendicular milling. Finally, we consider using FIB with other ions, in particular, with xenon, since it is far less reactive and also heavier than Ga. These two characteristics may enable fabrication of devices that are better, larger, and more suitable for transport measurements.”
But as we pointed out in A5 above, this is (a) a Conference paper reporting the work in progress, and (b) making negative results public can be of interest and value to the research community, preventing future mistakes and erroneous conclusions, and saving time, effort, and resources. Note that we used the standard FIB protocols that many other users at the same and different national user facilities have been following. This means they must have had the same issues, perhaps without realizing it. Once they read this, they may be compelled to revisit their earlier interpretations and claims.
C3. More details about the original bulk sample and its fabrication process should be provided. Where is the bulk LSCO sample purchased and what is the quality of the original sample before fabrication?
A3. We added details about the bulk crystal at line 168. The new text now reads: “A high-quality bulk LSCO single crystal with x = 0.07 was grown using the floating-zone technique at Brookhaven National Laboratory.”
C4. On Page 4, the authors wrote “We successfully optimized the contact deposition parameters to avoid the formation of Schottky barriers that could affect the transport measurements on the LSCO Hall bar devices.” What are the deposition parameters and how bad the Schottky barriers look like? Data should be provided to support this successful optimization.
A4. We added the optimized deposition parameters for Pt contacts in line 144-145. The new text reads: “We successfully optimized the contact deposition parameters by reducing the ion-beam energy down to 16 keV, as well as the beam current to 12 pA, to minimize the formation of Schottky barriers that could affect the transport measurements on the LSCO Hall bar devices”.
We observed a strong frequency dependence of the resistance in some of our devices – which signals the presence of a Schottky barrier instead of an ohmic contact - before Pt deposition parameters were tuned to improve the quality of the ohmic contact. To clarify the importance of this optimization, we added a new figure 4, showing the frequency dependence of the resistance before and after Pt deposition parameter optimization. We also changed the text accordingly, so that the new paragraph (lines 143-154) now reads: “We successfully optimized the contact deposition parameters by reducing the ion-beam energy down to 16 kV, as well as the beam current to 12 pA, to minimize the formation of Schottky barriers that could affect the transport measurements on the LSCO Hall bar devices. In Figure 4, we show the resistance versus frequency dependence for two different devices, before and after optimization of parameters of Pt deposition. In Figure 4a, we show the data from a device that was not patterned (and discarded as not useful for further analysis), while in Figure 4b we show the resistance versus frequency data for the sample shown in Figure 1b. In this case the resistance does not depend upon the frequency up to about 1 kHz, confirming the presence of an ohmic contact. However, our efforts to minimize Pt contamination so far yielded only marginal improvements, in the sense that regardless of this optimization we did not observe a complete superconducting transition.
C5. Given that the authors suspect Ga contamination is a main factor that suppresses the superconducting transition, the authors may want to consider FIB with other ions. People have explored FIB with ions other than Ga (for example, Liu, J., Niu, R., Gu, J., Cabral, M., Song, M. and Liao, X., 2020. Effect of ion irradiation introduced by focused ion-beam milling on the mechanical behaviour of sub-micron-sized samples. Scientific reports, 10(1), pp.1-8.)
A5. Our considerations on using different FIB sources are included among potential alternative methods on line 206. The new text reads: “Finally, we consider using FIB with other ions, as described for instance in [14], in particular, with xenon, since it is far less reactive and also heavier than Ga. These two characteristics may enable fabrication of devices that are better, larger, and more suitable for transport measurements.” We also included the reference (Liu et al, Sci. Rep. 2020) suggested by the Reviewer as reference 14 in the revised version of the manuscript.
Round 2
Reviewer 1 Report
In the revised manuscript, the authors have addressed my previous questions properly. On top of that, a new section is added to provide the TEM measurement result along with a discussion of the TEM data in the conclusion. The new addition of measurement helps in providing a full picture of the material properties. I’d suggest a publication of this manuscript on Condensed Matter.
Reviewer 3 Report
This revised version has addressed the questions from the original submission. In particular, the authors now include a number of constructive suggestions to potentially overcome the challenges and describe the details of the parameter optimization process. The overall quality is improved. I agree that despite the negative results, this work is still useful to other researchers in the field. Based on the reasons above, I would support its publication in Condensed Matter.